# OpenReview forum: "Robust Prompt Optimization for Defending Language Models Against Jailbreaking Attacks"
_NeurIPS.cc/2024/Conference — NeurIPS 2024 spotlight_

### Official Review · Reviewer_EVpE · 2024-07-05

**Soundness:** 3
**Presentation:** 3
**Contribution:** 3
**Rating:** 8
**Confidence:** 5

**Summary:**

This paper introduces a new approach called Robust Prompt Optimization (RPO) for defending large language models (LLMs) against jailbreaking attacks. The key contributions are:

1. Formalizing a minimax optimization objective for ensuring safe LLM outputs under a realistic threat model involving various attacks and adaptive adversaries.

2. Proposing the RPO algorithm, which directly optimizes for the defense objective using principled attack selection and discrete optimization.

3. Developing an easily deployable suffix-based defense that achieves state-of-the-art performance in protecting LLMs against jailbreaks on benchmark datasets.

The RPO method works by optimizing a set of "trigger tokens" that enforce safe outputs even under adversarial attacks. The authors evaluate RPO on recent red-teaming benchmarks and show it significantly reduces attack success rates on models like GPT-4 and LLaMA-2.

Key advantages of RPO include its negligible inference cost, minimal impact on benign prompts, and ability to transfer to black-box models and unknown attacks. The paper provides both theoretical analysis and experimental results demonstrating RPO's effectiveness as a robust, universal defense against various jailbreaking techniques.

**Strengths:**

Originality:
- Proposes the first formal optimization objective for defending language models against jailbreaking attacks, incorporating the adversary directly into the defensive objective. This is a novel formulation of the problem.
- Introduces Robust Prompt Optimization (RPO), a new algorithm to optimize for this defensive objective using a combination of attack selection and discrete optimization.

Quality:
- Provides theoretical analysis showing that optimizing their proposed objective is guaranteed to improve robustness, even on unseen instructions and attacks. This gives a solid theoretical grounding.
- Conducts extensive empirical evaluation on recent benchmarks (JailbreakBench and HarmBench), demonstrating state-of-the-art performance in reducing attack success rates.
- Shows transferability of the defense to black-box models like GPT-4 and resistance to adaptive attacks, indicating the approach is robust.

Clarity:
- The paper is generally well-structured and clearly written.
- Key ideas and contributions are summarized concisely in the introduction.
- The methodology is explained step-by-step with supporting equations and an algorithm description.

Significance:
- Addresses an important problem in AI safety - defending large language models against jailbreaking attacks that could lead to harmful outputs.
- Achieves state-of-the-art results on reducing attack success rates (down to 6% on GPT-4 and 0% on Llama-2).
- The proposed defense is lightweight and easily deployable as a suffix, making it practical for real-world implementation.
- The approach is model-agnostic and transfers well to different LLMs, including closed-source models, increasing its potential impact.

Overall, this paper makes significant contributions in formulating and addressing the challenge of defending LLMs against jailbreaking attacks. The combination of theoretical grounding, novel algorithmic approach, and strong empirical results on challenging benchmarks makes this work quite impactful for the field of AI safety and robustness.

**Weaknesses:**

1. Limited discussion of computational costs:
   - The paper doesn't provide details on the computational resources required for RPO optimization.
   - It's unclear how long it takes to generate the defensive suffix or how this scales with different model sizes.
   - Actionable improvement: Include a section on computational requirements, comparing RPO's runtime and resource usage to existing defenses and baseline LLM inference.

2. Lack of ablation studies:
   - The paper doesn't explore the impact of different components of RPO (e.g., attack selection frequency, batch size, number of iterations).
   - Actionable improvement: Conduct ablation studies to show how each component contributes to the overall performance and to guide practitioners in tuning these hyperparameters.

3. Limited exploration of potential negative impacts:
   - While the paper focuses on defending against harmful outputs, it doesn't discuss potential unintended consequences of the defense mechanism.
   - For instance, could RPO inadvertently block legitimate but sensitive queries?
   - Actionable improvement: Include a section on potential limitations and negative impacts, with empirical analysis on false positive rates for benign but sensitive queries.

4. Insufficient comparison to other optimization-based defenses:
   - While the paper compares to some existing defenses, it doesn't thoroughly compare to other optimization-based approaches in adversarial robustness literature.
   - Actionable improvement: Include comparisons to adversarial training methods adapted for language models, such as those proposed by Ziegler et al. (2022) in "Adversarial Training for High-Stakes Reliability".

5. Limited exploration of transfer learning:
   - While the paper shows transfer to GPT-4, it doesn't explore how well the defense transfers between models of different sizes or architectures.
   - Actionable improvement: Conduct experiments on transfer learning between models of varying sizes (e.g., from smaller to larger models) and different architectures (e.g., from decoder-only to encoder-decoder models).

6. Lack of human evaluation:
   - The paper relies primarily on automated metrics for evaluation.
   - It's unclear how the defended model's outputs are perceived by human users in terms of safety and quality.
   - Actionable improvement: Conduct a human evaluation study to assess the perceived safety and quality of outputs from models with and without RPO defense.

7. Limited discussion on the choice of loss function:
   - The paper uses log probability as the loss function but doesn't justify this choice or explore alternatives.
   - Actionable improvement: Provide a justification for the chosen loss function and experiment with alternative loss functions (e.g., KL-divergence, earth mover's distance) to see if they yield better results.

8. Insufficient analysis of the learned defensive suffixes:
   - The paper doesn't provide an in-depth analysis of the structure or content of the learned defensive suffixes.
   - Actionable improvement: Include a section analyzing the learned suffixes, perhaps using interpretability techniques to understand what patterns the defense is learning.

9. Limited exploration of multi-turn interactions:
   - The paper focuses on single-turn interactions, but many real-world scenarios involve multi-turn dialogues.
   - Actionable improvement: Extend the evaluation to multi-turn scenarios to assess how well the defense holds up over extended interactions.

10. Lack of discussion on potential adaptive attacks:
    - While the paper mentions resistance to adaptive attacks, it doesn't explore specific adaptive strategies an attacker might employ against RPO.
    - Actionable improvement: Include a section on potential adaptive attacks against RPO and empirically evaluate the defense's performance against these hypothetical attacks.

These specific improvements would strengthen the paper's contribution and provide more comprehensive insights into the proposed defense mechanism.

**Questions:**

1. Computational Resources and Scalability:
   Question: What are the computational requirements for RPO optimization? How does the runtime scale with model size and dataset size?
   Suggestion: Provide a detailed analysis of computational costs, including time and hardware requirements for different model sizes.

2. Hyperparameter Sensitivity:
   Question: How sensitive is RPO to its hyperparameters (e.g., attack selection frequency, batch size, number of iterations)?
   Suggestion: Conduct and present an ablation study showing the impact of different hyperparameter choices on the defense's effectiveness.

3. False Positive Analysis:
   Question: Does RPO inadvertently block legitimate but sensitive queries? What is the false positive rate?
   Suggestion: Perform an analysis on a set of benign but potentially sensitive queries to assess any unintended blocking.

4. Comparison to Adversarial Training:
   Question: How does RPO compare to adversarial training methods adapted for language models?
   Suggestion: Include a direct comparison with adversarial training approaches, particularly those designed for language models.

5. Transfer Learning Capabilities:
   Question: How well does the RPO defense transfer between models of different sizes or architectures?
   Suggestion: Conduct experiments showing transfer performance between various model sizes and architectures.

6. Human Evaluation:
   Question: How do human evaluators perceive the safety and quality of outputs from RPO-defended models compared to undefended models?
   Suggestion: Conduct a human evaluation study and present the results.

7. Loss Function Choice:
   Question: Why was log probability chosen as the loss function? Have alternative loss functions been considered?
   Suggestion: Provide justification for the chosen loss function and experiment with alternatives like KL-divergence or earth mover's distance.

8. Analysis of Learned Suffixes:
   Question: What patterns or structures are present in the learned defensive suffixes?
   Suggestion: Perform an in-depth analysis of the learned suffixes, possibly using interpretability techniques.

9. Multi-turn Interactions:
   Question: How does RPO perform in multi-turn dialogue scenarios?
   Suggestion: Extend the evaluation to include multi-turn interactions to assess the defense's effectiveness over extended conversations.

10. Adaptive Attacks:
    Question: What specific adaptive strategies might an attacker employ against RPO, and how does the defense perform against these?
    Suggestion: Outline potential adaptive attacks and empirically evaluate RPO's performance against them.

11. Impact on Model Performance:
    Question: Does the RPO defense impact the model's performance on non-adversarial tasks?
    Suggestion: Evaluate the defended model on standard language modeling benchmarks to assess any potential degradation in general performance.

12. Generalization to Other Types of Attacks:
    Question: How well does RPO generalize to types of attacks not seen during training?
    Suggestion: Test the defense against a held-out set of novel attack types not used during optimization.

13. Ethical Considerations:
    Question: Are there any potential ethical issues or misuse scenarios associated with deploying RPO in real-world applications?
    Suggestion: Include a discussion on the ethical implications and potential misuse of the technology.

14. Integration with Existing Systems:
    Question: How easily can RPO be integrated into existing language model deployment pipelines?
    Suggestion: Provide guidelines or a case study on integrating RPO into a typical LLM deployment setup.

15. Longevity of the Defense:
    Question: How long is the RPO defense expected to remain effective as new attack methods are developed?
    Suggestion: Discuss the expected longevity of the defense and propose strategies for keeping it up-to-date with evolving attacks.

These questions and suggestions aim to address key aspects of the paper that could benefit from further clarification or exploration, potentially changing opinions on the work's impact and completeness.

**Limitations:**

Yes, the authors adequately addressed the limitations.

---

> ### Author Rebuttal · Authors · 2024-08-07
>
> We thank the reviewer for their insightful feedback and positive review. We're pleased that the reviewer found our work to have *significant contributions in formulating and addressing the challenge of defending LLMs against jailbreaking attacks* and appreciated our *combination of theoretical grounding, novel algorithmic approach, and strong empirical results.* We address concerns and suggestions below:
>
> **Weaknesses**
>
> *Computational costs*: Optimizing the RPO suffix takes about 4-5 hours on a single A100 GPU for 500 iterations, each taking 1-2 minutes. We observe convergence after ~100 iterations. This is approximately 8x cheaper than optimizing a GCG attack string. We will add these details to the paper.
>
> *Ablation studies*: Thank you for this suggestion. While a full ablation study would be valuable, it's computationally infeasible within the rebuttal period, given our resources. However, we did observe that increasing the number of iterations and batch size generally improved performance, with diminishing returns after 100 iterations. We'll include these observations in the paper and conduct an analysis.
>
> *Potential negative impacts*: In our experiments, we found that RPO had minimal impact on benign queries, with a slight performance reduction on MT-Bench (9.32 vs 9.20 on GPT-4), only on very short queries where the model might get confused by the suffix. We'll expand our discussion in Section 4.3.
>
> *Comparison to other optimization-based defenses*: While a direct comparison to adversarial training methods would be interesting, it's challenging due to the significant differences in approach and computational requirements. However, we'll expand our discussion in the related work section to highlight these differences and potential complementarities.
>
> *Transfer learning*: We agree this is an interesting direction. Our current results in Table 1 demonstrate transfer across different model sizes and architectures (e.g., from Llama-2 to GPT-4 and Vicuna).
>
> *Human evaluation*: Thank you for the suggestion. While a full human evaluation study is beyond our current resources, we agree it would be valuable. We'll discuss this limitation in the paper.
>
> *Loss function choice*: We chose log probability as it's standard in language modeling and aligns well with our objective. We'll add this justification to the paper.
>
> *Analysis of learned suffixes*: We'll add a brief analysis of common patterns in the learned suffixes to the paper, though a full interpretability study is beyond our current scope.
>
> *Multi-turn interactions*: Our current work focuses on single-turn interactions, which are the standard setting in LLM security. All attack baselines were evaluated using single-turn instructions in their original settings, which we match for our defense setting. Extending to multi-turn scenarios is an important direction for future work, which we'll highlight in the paper.
>
> *Adaptive attacks*: We discuss adaptive attacks in Section 4.3 and Table 3. We designed adaptive versions of GCG and PAIR and found that RPO retains its high robustness, and has stronger adaptive robustness than defense baselines.
>
> **Questions**
>
> *Impact on model performance*: We analyze the trade-off between robustness and general performance in Table 4. We find that there is only a minor trade off between robustness and model performance, typically for very short queries where the model might get confused by the suffix. This results in a slight reduction on MT-Bench. On MMLU, we do not observe a performance difference for most models. Optimizing for semantically meaningful defense strings could fully mitigate this, which we leave to future work.
>
> *Generalization to unseen attacks*: Table 2 demonstrates RPO's generalization to unseen attacks on HarmBench. RPO reduces ASR across all six attacks, including four not seen during optimization. For example, on Vicuna-13B, RPO reduces ASR from 65.9% to 59.5% on AutoDAN and from 53.6% to 37.2% on TAP. This robust generalization showcases RPO's effectiveness against a wide range of jailbreaking techniques.
>
> *Ethical considerations*: Section 5 provides a discussion of ethical implications. We acknowledge that proposing our defense may lead to the development of stronger attacks. However, we believe the benefits of improved LLM safety outweigh this risk.
>
> *Integration with existing systems*: RPO is designed for easy integration into existing LLM deployment pipelines. The optimized suffix (20 tokens) is appended to the user input as part of the system prompt during inference. This requires minimal changes to existing infrastructure and incurs negligible computational overhead compared to baseline defenses like SmoothLLM.
>
> *Longevity of the defense*: While the evolving nature of attacks poses challenges, RPO's design allows for easy updating. The suffix can be periodically re-optimized on new attack types to maintain effectiveness. We conduct an analysis where we add a new attack, TAP, to optimization below.
>
> | Model       | GCG  | AutoDan | PAIR | TAP | Few-Shot | PAP | Average |
> |-------------|------|---------|------|-----|----------|-----|---------|
> | Llama-2-7B  | 31.9 | 0.0     | 9.4  | 9.1 | 5.0      | 2.7 | 9.7     |
> | + RPO  (3 attacks)     | 6.7  | 0.0     | 5.0  | 7.8 | 0.0      | 0.0 | 3.2     |
> | + RPO  (4 attacks inc TAP)     | 6.9  | 0.0     | 4.7  | 5.2 | 0.0      | 0.0 | 2.8     |
>
> We find that RPO can indeed be updated to new attacks, further improving the robustness of TAP by including it in the optimization. This results in higher average robustness after adding a new attack. In practice, RPO generalizes well to unseen attacks, so it is not mandatory to keep updating the defense, although this can indeed improve robustness further.
>
> We hope this addresses your concerns. We are happy to answer further questions in the discussion phase.

---

> ### Author Response · Authors · 2024-08-12
> **Follow up to reviewer**
>
> Dear Reviewer EVpE,
>
> We are thankful for your review. With the rebuttal deadline drawing near, please inform us if your concerns have been properly addressed. We are ready to offer further clarification.
>
> Best regards,
>
> The Authors

---

### Official Review · Reviewer_YYJE · 2024-07-12

**Soundness:** 3
**Presentation:** 3
**Contribution:** 3
**Rating:** 7
**Confidence:** 2

**Summary:**

This paper introduces Robust Prompt Optimization (RPO), a novel method for defending LLM against jailbreaking attacks, which manipulate prompts to induce harmful behavior. Inspired by the Adversarial Training, RPO optimizes a suffix for the LLM prompt, ensuring safe responses even when the input is modified by an attacker. The RPO method contains two steps, jailbreaking prompt selection and discrete optimzation, which is also based on a complete theoretical proof. RPO demonstrates significant improvements in attack success rate (ASR) on various LLMs, including GPT-4, and is transferable to unknown attacks, making it a robust and versatile defense mechanism for LLMs.

**Strengths:**

**Good Innovation**: Optimization-based methods are widely used for optimizing jailbreak suffixes, but this paper innovatively applies it to the reinforcement of security alignment, introducing a new strategy for jailbreak defense.

**Solid Theoretical Foundation**: The paper conducts a detailed theoretical analysis of RPO (Reinforced Protection Optimization).

**Elaborate Empirical Evaluation**: The effectiveness of the method is demonstrated through extensive empirical experiments.

**Weaknesses:**

**Transferability Discussion**: While RPO shows promising transferability, which is very interesting, the paper does not seem to discuss this aspect sufficiently. I recommend that the authors further explore and elaborate on why RPO exhibits good transferability.

**Efficiency Concerns**: Section 4.3 discusses the method only adds 20 additional tokens, which does not significantly increase the cost of inference for benign prompts. However, there appears to be a lack of discussion on the efficiency of the optimization process itself. It is well-known that token-level optimization, typified by methods like GCG, is often criticized for its high optimization costs. Thus, I am concerned about the optimization efficiency of RPO.

**Questions:**

1.How many rounds of optimization are required for a successful optimization of RPO, and how much time does each round typically take?

2.Why do the optimized defense suffixes exhibit good transferability across different models?

**Limitations:**

The paper thoroughly discusses the existing limitations.

---

> ### Author Rebuttal · Authors · 2024-08-07
>
> We thank the reviewer for the insightful and positive review. We are glad the reviewer found our use of optimization for defense innovative, theoretical analysis solid, and empirical evaluation elaborate and extensive.
>
> > *Transferrability discussion*
>
> We are glad the reviewer found the transferability of RPO suffixes interesting. This property can be attributed to the similarities between LLMs, all trained with similar autoregressive objectives and internet-scale datasets. This is supported by the fact that many LLMs from different families, such as Llama and GPT, now have similar capabilities. We will add a discussion about this in the revised version.
>
>
> > *Effiency concerns*
>
> Optimizing the RPO suffix indeed incurs a computational cost, but it is less than the cost of GCG. Our experiments find that optimizing this suffix is around 8x cheaper than optimizing a GCG attack string since models already tend to refuse harmful instructions. We also optimize this on a single A100 GPU, which takes about 4-5 hours for 500 iterations. Each iteration takes 1-2 minutes, and we observe convergence after around 100 iterations. Further reducing this cost could be an interesting avenue for future work as explored in recent attack optimization methods [1]. We will add a more detailed discussion of this in the revised version.
>
> We hope our response answered the reviewer's remaining concerns. We are happy to answer further questions in the discussion phase.
>
> [1] Sadasivan, V.S., Saha, S., Sriramanan, G., Kattakinda, P., Chegini, A.M., & Feizi, S. (2024). Fast Adversarial Attacks on Language Models In One GPU Minute. ArXiv, abs/2402.15570.

---

> ### Author Response · Authors · 2024-08-12
> **Follow up to reviewer**
>
> Dear Reviewer YYJE,
>
> We are thankful for your review. With the rebuttal deadline drawing near, please inform us if your concerns have been properly addressed. We are ready to offer further clarification.
>
> Best regards,
>
> The Authors

---

### Official Review · Reviewer_yUwN · 2024-07-12

**Soundness:** 4
**Presentation:** 4
**Contribution:** 3
**Rating:** 6
**Confidence:** 5

**Summary:**

The paper proposes a novel method, Robust Prompt Optimization (RPO), to enhance the robustness of large language models (LLMs) against jailbreaking attacks. Existing defenses, which operate during pre-training or fine-tuning stages, incur high computational costs. The proposed RPO method optimizes a lightweight suffix at the inference stage, incorporating the adversary into the defensive objective. Theoretical and experimental results demonstrate that RPO reduces the attack success rate on GPT-4 and Llama-2 models, setting a new state-of-the-art in defense against jailbreaking attacks.

**Strengths:**

1. The proposed method demonstrates substantial improvements in reducing attack success rates, outperforming existing defense mechanisms.
2. Unlike other methods that operate during pre-training or fine-tuning stages, RPO is computationally efficient, operating during the inference stage without significant overhead.
3. The authors provide a thorough theoretical analysis of the defense method.

**Weaknesses:**

1. The method can not address other failure modes such as deception and malicious code generation, limiting its applicability.
2. How does the proposed RPO method handle the evolution of attack strategies over time, and how frequently would the defensive suffix need updating? The authors should have provided more analysis.
3. What are the potential trade-offs between robustness and model performance when applying RPO to different LLM architectures? The authors should have provided more analysis.

**Questions:**

Please refer to Weaknesses for details.

**Limitations:**

Yes.

---

> ### Author Rebuttal · Authors · 2024-08-07
>
> Thank you for your thorough review and positive assessment of our work. We greatly appreciate the reviewer's recognition of our paper's substantial improvements in reducing attack success rates and its computational efficiency. We're pleased that the reviewer found our theoretical analysis thorough. We address concerns and questions in detail below:
>
> > *Handling other failure modes*
>
> Indeed, our paper does not focus on other failure modes, such as deception and malicious code generation, but on harmful generation. We acknowledge this limitation in Section 5 of our paper. It is worth noting that these failure modes are very nascent and not well-studied, lacking standardized definitions and benchmarks, but they could be an interesting direction for future work.
>
> > *Evolution of attack strategies and updating the defensive suffix*
>
> The RPO method is designed to be adaptive and can be periodically reoptimized to account for new attack strategies. Indeed, initial versions of RPO were only optimized on GCG and had limited transferability to semantically meaningful attack prompts in PAIR or JBC. To further analyze this, we optimize a new RPO suffix on TAP, a newer attack, in addition to GCG, PAIR, and JBC. The results on HarmBench are provided below.
>
> | Model       | GCG  | AutoDan | PAIR | TAP | Few-Shot | PAP | Average |
> |-------------|------|---------|------|-----|----------|-----|---------|
> | Llama-2-7B  | 31.9 | 0.0     | 9.4  | 9.1 | 5.0      | 2.7 | 9.7     |
> | + RPO  (3 attacks)     | 6.7  | 0.0     | 5.0  | 7.8 | 0.0      | 0.0 | 3.2     |
> | + RPO  (4 attacks inc TAP)     | 6.9  | 0.0     | 4.7  | 5.2 | 0.0      | 0.0 | 2.8     |
>
> We find that RPO is indeed can be directly updated to new attacks, further improving the robustness of TAP by including it in the optimization. This also improves PAIR robustness, likely due to the similarity between the optimized prompts. This results in higher average robustness after adding a new attack. However, we also observe a slight reduction in GCG robustness, which has very different attack prompts, suggesting some interference between some attacks while benefits for others. We will conduct a more detailed analysis for the revised version of the paper.
>
> In addition, while we observe optimizing directly on an attack will improve robustness directly, RPO can also transfer well to unseen attacks, making it not mandatory to update the defense whenever there is a new attack.
>
> > *Trade-offs between robustness and model performance for different LLM architectures*
>
> We appreciate you highlighting this important aspect. We analyze the trade-off between robustness and general performance in Table 4 (also attached below). We find that there is only a minor trade-off between robustness and model performance, typically for very short queries where the model might get confused by the suffix. This results in a slight reduction on MT-Bench. More capable models such as GPT-4 are more robust and have a more minor performance reduction. On MMLU, we do not observe a performance difference for most models. Optimizing for semantically meaningful defense strings could mitigate this, which we leave to future work.
>
> | Model       | Method | MT-Bench | MMLU |
> |-------------|--------|----------|------|
> | Vicuna-13B  | Base   | 6.57     | 0.50 |
> |             | RPO    | 5.96     | 0.49 |
> | Llama-2-7B  | Base   | 6.18     | 0.46 |
> |             | RPO    | 6.05     | 0.46 |
> | GPT-3.5     | Base   | 8.32     | 0.68 |
> |             | RPO    | 7.81     | 0.66 |
> | GPT-4       | Base   | 9.32     | 0.85 |
> |             | RPO    | 9.20     | 0.85 |
>
> Thank you again for your insightful comments and questions. We are happy to answer additional concerns or questions in the discussion phase.

---

> ### Author Response · Authors · 2024-08-12
> **Follow up to reviewer**
>
> Dear Reviewer yUwN,
>
> We are thankful for your review. With the rebuttal deadline drawing near, please inform us if your concerns have been properly addressed. We are ready to offer further clarification.
>
> Best regards,
>
> The Authors

---

### Official Review · Reviewer_Md6V · 2024-07-13

**Soundness:** 3
**Presentation:** 3
**Contribution:** 3
**Rating:** 7
**Confidence:** 4

**Summary:**

In this paper, the authors proposes a new defense, called Robust Prompt Optimization(RPO) to defend the jailbeak attack. It optimizes a secure suffix with min-max optimization. Experiments reveal that it can defend multiple existing attacks

**Strengths:**

1 This paper is well written.

2 The authors give theoretical proof to explain the working dynamics of RPO.

3 The proposed method is simple and easy to understand.

**Weaknesses:**

1 RPO obtains high ASR against the JBC attack on Vicuna which is much worse than the Rephrasing defense.

2 Although RPO can save a lot of computation cost than other methods during the inference time. The cost to generated the suffix can not be simply ignored. From my experimence, it usually needs multiple hours even on a single A100 GPUs. Regarding this point, self-reminder [1] and ICD [2] might be a better choice.

3 In Table 1, authors only report results on two open-source and closed-source models. More experiments are needed to fully investigate the capability of RPO, such as Vicuna-7B, Llama-13B,  QWen-7B and claude.

4 Although in the related work section, the authors list a lot of current defenses. However, in Table 1, they compare RPO with only a few. I would suggest authors compared DPO with stronger defenses such as [3] , [4] and [5].

**Questions:**

See weakness.

**Limitations:**

Authors well addresses the limitations of the proposed method.

[1] Defending chatgpt against jailbreak attack via self-reminder

[2] Jailbreak and guard aligned language models with only few in-context demonstrations

[3]  Jailbreak and Guard Aligned Language Models with Only Few In-Context Demonstrations

[4]  Defending large language models against jailbreaking attacks through goal prioritization.

[5]  Rain: Your language models can align themselves without finetuning

---

> ### Author Rebuttal · Authors · 2024-08-07
>
> Thank you for the insightful and helpful review. We are glad the reviewer found the paper well-written, theoretical results meaningful, and method easy to understand. We address the concerns below.
>
> > *RPO obtains high ASR against the JBC attack on Vicuna which is much worse than the Rephrasing defense.*
>
> 1. While RPO indeed performs worse than rephrasing on Vicuna on the JailbreakChat attack, we note that this is the only setting where RPO is not the strongest defense. On Llama-2, GPT-3.5, and GPT-4, RPO outperforms all defenses. These are also more important and widely used models than Vicuna. On the more difficult, optimization-based attacks GCG and PAIR, RPO outperforms baseline defenses on Vicuna.
>
> 2. Vicuna is an older model that has no safety training. This may result in fundamentally different behavior when presented with jailbreaking attempts, making it more susceptible to certain attacks and less responsive to our defense mechanism. Vicuna, based on an earlier Llama model, may have different attention patterns and token representations compared to Llama-2, on which RPO was optimized. We will add this discussion in the revised version.
>
> > *The cost to generated the suffix can not be simply ignored.*
>
> Optimizing the RPO suffix indeed incurs a computational cost. Our experiments find that optimizing this suffix is around 8x cheaper than optimizing a GCG attack string since models already tend to refuse harmful instructions. We also optimize this on a single A100 GPU for 4-5 hours. We also open-source our optimized RPO suffixes, so users do not need to do this themselves. Our new experimental results below demonstrate RPO outperforms defenses with similar inference costs, such as Self-Reminder and Few-Shot Examples.
>
> > *More experiments are needed to fully investigate the capability of RPO*
>
> We conducted our experiments on four models: Llama-2-7B, Vicuna-13B, GPT-3.5-Turbo, and GPT-4 since these are the models represented in JailbreakBench [1]. As suggested, we have conducted additional evaluations on the Qwen-7B and Llama2-13B models. Due to the cost of generating new attack prompts for each instruction for each model, we will evaluate Vicuna-7B and Claude for the revised paper. The results further demonstrate the effectiveness and transferability of RPO across different model architectures and sizes.
>
> | Method (Qwen-7B)  | PAIR | GCG  | JBC  |
> |--------------------|------|------|------|
> | Base               | 0.68 | 0.11 | 0.58 |
> | Perplexity Filter  | 0.66 | 0.0  | 0.58 |
> | SmoothLLM          | 0.36 | 0.02 | 0.44  |
> | Few-Shot           | 0.16 | 0.01 | 0.50 |
> | RPO                | 0.04 | 0.0  | 0.45 |
>
> For Qwen-7B, RPO significantly outperforms all other methods on the PAIR attack, reducing the attack success rate from 68% (base model) to just 4%. This is a substantial improvement over the next best defense, Few-Shot, which achieves 16%. On the GCG attack, RPO matches the best performance of 0% attack success rate. On JBC, RPO is competitive with SmoothLLM (but outperforms it significantly on PAIR and GCG).
>
> | Method (Llama2-13B) | PAIR | GCG | JBC  |
> |---------------------|------|-----|------|
> | Base                | 0.02 | 0.0 | 0.01 |
> | Perplexity Filter   | 0.02 | 0.0 | 0.01 |
> | SmoothLLM           | 0.01 | 0.0 | 0.0  |
> | Few-Shot            | 0.01 | 0.0 | 0.01 |
> | RPO                 | 0.01 | 0.0 | 0.00 |
>
> Similar to Llama-2-7B, Llama-2-13B already has high robustness to all three attacks. Using RPO similarly outperforms baseline defenses across attacks, achieving perfect robustness to GCG and JBC and reducing PAIR ASR. SmoothLLM is the only defense competitive to RPO in this setting, but it requires twice as many inference computations and is less effective on other models. We will add these results to the revised version and similar evaluations on Claude and Vicuna-7B.
>
> > *I would suggest authors compared DPO with stronger defenses*
>
> We appreciate the reviewer's suggestion to compare RPO with additional strong defenses. We have conducted further experiments, comparing RPO with self-reminder and few-shot examples (RAIN is a weaker defense that has a very high GCG ASR on Vicuna (38%) [2], so we do not evaluate it). The results for GPT-4 are presented in the table below.
>
> | Method (GPT-4) | PAIR | GCG  | JBC  |
> |-----------------|------|------|------|
> | Base                | 0.50 | 0.01 | 0.0 |
> | Perplexity Filter | 0.43 | 0.0  | 0.0  |
> | SmoothLLM      | 0.25 | 0.03 | 0.0  |
> | Rephrasing     | 0.35 | 0.01 | 0.01 |
> | Self-reminder  | 0.16 | 0.0  | 0.0  |
> | Few-Shot       | 0.10 | 0.0  | 0.0  |
> | RPO            | 0.06 | 0.0  | 0.0  |
>
> These new results demonstrate that RPO consistently outperforms or matches the performance of all compared defenses across different attack types:
>
> PAIR Attack: RPO achieves the lowest attack success rate of 6%, significantly improving upon Self-reminder (16%) and Few-Shot (10%).
> GCG Attack: RPO matches the perfect defense (0% attack success rate) achieved by Perplexity Filter, Self-reminder, and Few-Shot, while outperforming SmoothLLM and Rephrasing.
> JBC Attack: RPO maintains the 0% attack success rate achieved by most defenses, matching the performance of baselines.
>
> These results highlight that RPO not only compares favorably to the defenses in our original submission but also outperforms or matches the effectiveness of additional strong defenses suggested by the reviewer. We will incorporate these additional comparisons and analyses into the revised paper and add the results of new baseline defenses on the other models we evaluated.
>
> We hope our response has addressed the reviewer's concerns. We are happy to answer further questions in the discussion phase.
>
> [1] Chao, P., Debenedetti et al (2024). JailbreakBench: An Open Robustness Benchmark for Jailbreaking Large Language Models. ArXiv, abs/2404.01318.
>
> [2] Li, Y., Wei et al (2023). RAIN: Your Language Models Can Align Themselves without Finetuning. ArXiv, abs/2309.07124.

---

> ### Author Response · Authors · 2024-08-12
> **Follow-up to reviewer**
>
> Dear Reviewer Md6V,
>
> We are thankful for your review. With the rebuttal deadline drawing near, please inform us if your concerns have been properly addressed. We are ready to offer further clarification.
>
> Best regards,
>
> The Authors

---

> ### Comment · Reviewer_Md6V · 2024-08-13
>
> Dear authors,
>
> I am happy that you address most of my concern.
>
> 1 However, considering defending jailbreak attacks from prompt tuning view is not a full new direction. For example, related works [1] and [2] study the same problem and propose the corresponding methods. I list them here is not doubting the novelty of this paper. However, discussing with them might help readers understand the importance of the prompt-based defenses better.
>
>
> 2 Notice that you release the code. How many iterations do RPO need for its convergence?
>
>
> [1] Fight Back Against Jailbreaking via Prompt Adversarial Tuning
>
> [2] On prompt-driven safeguarding for large language models

---

> ### Author Response · Authors · 2024-08-13
> **Response to reviewer**
>
> We thank the reviewer for the response and are glad most concerns are addressed.
>
> > *Discussing related work*
>
> We are glad the reviewer has brought these works to our attention and agree they should be discussed.
> 1. PAT [1] is concurrent to our work, and explores a similar optimization objective. However, PAT is only optimized on the GCG attack, which may limit its transferability to newer attacks.
> 2. DRO [2] also explores prompt-based defenses for LMs but optimizes a soft prompt in embedding space. This prevents DRO from transferring to closed-source models. In addition, the DRO objective is only a minimization objective, not a min-min optimization objective like RPO, which incorporates the adversary into optimization. This makes RPO more adaptive to jailbreaking attacks.
>
> We will include baseline comparisons and a more detailed discussion in the revised version.
>
> >*How many iterations do RPO need for its convergence?*
>
> In practice, we find that RPO requires around 100 iterations to converge on a single A100 GPU (around an hour), and we optimize our suffixes for 500 iterations, which takes 4-5 hours. In the revised version, we will conduct a more detailed analysis of performance over time.
>
> We are happy to further clarify, and hope the reviewer will raise the score if all concerns have been addressed.

---

> > ### Comment · Reviewer_Md6V · 2024-08-14
> >
> > Thank you for your response. However. from my view, PAT and DRO both incorporate benign prompts into their formulation, which makes them achieve lower FPR (False positive rate) on detecting harmless prompts. In RPO, I only see optimization on harmful input. Do you have any suggestions for future improvement?

---

> > > ### Author Response · Authors · 2024-08-14
> > > **Response to reviewer**
> > >
> > > Thank you for pointing this out. PAT and DRO indeed also optimize on benign prompts. However, from the reported MT-Bench scores in [1], RPO is competitive with PAT on FPR. We provide the results here, where the MT-Bench scores are taken directly from [1].
> > >
> > > | Model       | Method | MT-Bench |
> > > |-------------|--------|----------|
> > > | Vicuna-13B  | Base   | 6.57     |
> > > |              | PAT    | 6.15     |
> > > |             | RPO    | 5.96     |
> > > | GPT-3.5     | Base   | 8.32     |
> > > |             | PAT    | 8.06     |
> > > |             | RPO    | 7.81     |
> > > | GPT-4       | Base   | 9.32     |
> > > |             | PAT    | 8.77    |
> > > |             | RPO    | 9.20     |
> > >
> > > On weaker models such as Vicuna and GPT-3.5, RPO has a more significant effect on benign prompts than PAT, but on more capable models such as GPT-4, it has a smaller effect than PAT. This may be due to optimizing RPO on semantically meaningful harmful prompts. We will conduct a more detailed analysis and comparison in the revised version.
> > >
> > > Including benign prompts in the optimization of RPO is an interesting suggestion that may further reduce benign impact. We will try this improvement as an ablation in the revised version. For future work, modifying RPO to optimize semantically meaningful prompts, such as constraining candidate tokens to low perplexity ones, may also improve this.
> > >
> > >
> > > [1] Mo, Y., Wang, Y., Wei, Z., & Wang, Y. (2024). Fight Back Against Jailbreaking via Prompt Adversarial Tuning.

---

> > > > ### Comment · Reviewer_Md6V · 2024-08-14
> > > >
> > > > Thank you for your reply. I will increase my score to 7. Can not wait to see the change in the revised version of this paper.

---

> > > > > ### Author Response · Authors · 2024-08-14
> > > > >
> > > > > Thanks for appreciating our rebuttal and increasing the score! We will add all the new results and discussion in the revision.

---

### Author Rebuttal · Authors · 2024-08-07

Dear Reviewers and Area Chair,

We sincerely thank you for your thorough and insightful reviews of our paper on Robust Prompt Optimization (RPO). We appreciate the positive feedback and constructive suggestions that have helped improve our work.

We are pleased that the majority of reviewers were positive about the paper and that reviewers found our paper to have significant strengths. Reviewers noted that our paper is "well written" with a "simple and easy to understand" method, supported by "theoretical proof" and "thorough theoretical analysis." Our work was recognized for its "Good Innovation" in applying optimization to security alignment, with "Elaborate Empirical Evaluation" demonstrating "substantial improvements in reducing attack success rates." Importantly, it was highlighted that our work "Addresses an important problem in AI safety" and achieves "state-of-the-art results on reducing attack success rates."

In response to your feedback, we have:
- Conducted additional experiments on additional models (Qwen-7B and Llama-2-13B) to demonstrate RPO's broad effectiveness.
- Expanded comparisons with recent defense methods, self-reminder and few-shot examples.
- Provided more detailed analysis of RPO's computational requirements and optimization process.
- Elaborated on RPO's transferability and generalization to unknown attacks and conducted additional ablations on updating the defense string to new attacks

We believe these additions strengthen our paper significantly. We remain excited about RPO's potential impact on improving LLM safety and reliability, and we look forward to presenting a revised version incorporating these improvements.

Thank you again for your valuable feedback and for considering our work.

---

### Public Comment · ~Sungyoon_Lee1 · 2025-09-28

The algorithm says
$$A^\ast =\arg\min_{\mathcal{A}} \mathcal{L}\_j^{\text{adv}}\sum\_{1\leq o\leq m} (A_o(x^{(j)})).$$
So the loss is evaluated with the ``summed inputs'' $\sum\_{1\leq o\leq m} (A_o(x^{(j)}))$. How do we take the summation over discrete inputs? It seems weird.

I think it should be
$$A^\ast =\arg\min_{A\in\mathcal{A}} \sum\_{1\leq j\leq m}\mathcal{L}\_j^{\text{adv}} (A(x^{(j)}))$$
or something else.

---

### Decision · Program_Chairs · 2024-09-25

**Decision:**

Accept (spotlight)

**Comment:**

**Summary.** This work proposes a defense method to prevent jailbreaking attacks against LLMs. The technique, called Robust Prompt Optimization (RPO) optimizes a secure suffix at the inference stage (making it more lightweight than other competing methods) via min-max optimization, and demonstrates effectiveness against existing attacks on two state-of-the-art models.

**Positive aspects.** The contribution is substantial and well presented, as noted by all reviewers, and provides significant advancement to the field, bringing insights that are of high interest to the NeurIPS community. The method is supplied with theoretical proof to support the (extensive) empirical findings. The authors provide the code to reproduce results.

**Summary of rebuttal.** The authors have been very prepared to respond to the reviewers' questions, even with multiple interactions. Overall, the concerns were almost all addressed by them, at least the most important ones. The main concerns raised by the reviewers were:

1. Additional comparisons (Md6V, EVpE) [experiments provided]
2. More information on the computational cost and efficiency (Md6V, YYJE, EVpE) [experiments provided]
3. Applicability is limited (e.g., other failure modes are still effective) (yUwN) [expressed as limitation]
4. Can the method resist evolution of the attack along time and adaptive attacks (yUwN, EVpE) [partially addressed]
5. Trade off robustness vs model performance (yUwN)[experiments provided]
6. Clarifications on the transferability (YYJE, EVpE)[discussion added in the paper]
7. Ablation studies to understand the single components alone, choice of loss function (EVpE)[not addressed]
8. Exploration on potential negative impacts (EVpE) [discussion added]
9. Lack of human evaluation (EVpE) [only addressed as limitation in the paper]
10. Multi-turn interaction (EVpE)[highlighted as limitation in the paper]

**Final decision.** Given the importance of the topic, the scores for this paper, the quality of the work, and the productive rebuttal, I would recommend it as oral or spotlight.

I encourage the authors to include all material provided during the author response in the camera-ready version as it reveals additional insights and raise even further the interest for this paper.